# Discovery of Novel Spike Inhibitors against SARS-CoV-2 Infection

**DOI:** 10.3390/ijms25116105

**Published:** 2024-06-01

**Authors:** Li-Te Tai, Cheng-Yun Yeh, Yu-Jen Chang, Ju-Fang Liu, Kai-Cheng Hsu, Ju-Chien Cheng, Chih-Hao Lu

**Affiliations:** 1Industrial Development Graduate Program of College of Biological Science and Technology, National Yang Ming Chiao Tung University, Hsinchu 300193, Taiwan; 2Graduate Institute of Biomedical Sciences, China Medical University, Taichung 404333, Taiwan; 3The Ph.D. Program of Biotechnology and Biomedical Industry, China Medical University, Taichung 404333, Taiwan; 4School of Oral Hygiene, College of Oral Medicine, Taipei Medical University, Taipei 110301, Taiwan; 5Graduate Institute of Cancer Biology and Drug Discovery, Taipei Medical University, Taipei 110301, Taiwan; 6Department of Medical Laboratory Science and Biotechnology, China Medical University, Taichung 404333, Taiwan; 7Institute of Bioinformatics and Systems Biology, National Yang Ming Chiao Tung University, Hsinchu 300193, Taiwan; 8Department of Biological Science and Technology, National Yang Ming Chiao Tung University, Hsinchu 300193, Taiwan; 9Center for Intelligent Drug Systems and Smart Bio-Devices (IDS2B), National Yang Ming Chiao Tung University, Hsinchu 300193, Taiwan

**Keywords:** SARS-CoV-2, virtual screening, spike protein, entry inhibitors

## Abstract

Severe acute respiratory syndrome coronavirus-2 (SARS-CoV-2) is responsible for the current coronavirus disease pandemic. With the rapid evolution of variant strains, finding effective spike protein inhibitors is a logical and critical priority. Angiotensin-converting enzyme 2 (ACE2) has been identified as the functional receptor for SARS-CoV-2 viral entry, and thus related therapeutic approaches associated with the spike protein–ACE2 interaction show a high degree of feasibility for inhibiting viral infection. Our computer-aided drug design (CADD) method meticulously analyzed more than 260,000 compound records from the United States National Cancer Institute (NCI) database, to identify potential spike inhibitors. The spike protein receptor-binding domain (RBD) was chosen as the target protein for our virtual screening process. In cell-based validation, SARS-CoV-2 pseudovirus carrying a reporter gene was utilized to screen for effective compounds. Ultimately, compounds C2, C8, and C10 demonstrated significant antiviral activity against SARS-CoV-2, with estimated EC_50_ values of 8.8 μM, 6.7 μM, and 7.6 μM, respectively. Using the above compounds as templates, ten derivatives were generated and robust bioassay results revealed that C8.2 (EC_50_ = 5.9 μM) exhibited the strongest antiviral efficacy. Compounds C8.2 also displayed inhibitory activity against the Omicron variant, with an EC_50_ of 9.3 μM. Thus, the CADD method successfully discovered lead compounds binding to the spike protein RBD that are capable of inhibiting viral infection.

## 1. Introduction

The worldwide COVID-19 pandemic has deeply affected global public health, economies, and societies, causing several million fatalities [1,2,3]. Because the pandemic situation is not yet fully under control, the rapid evolution of variant strains and their potential resistance to current vaccines emphasizes the vital need to find effective spike protein inhibitors. Drug development primarily revolves around two main strategies: (1) preventing the virus from binding to host cell receptors, and (2) limiting the viral RNA synthesis and replication process by targeting critical viral enzymes. Currently, the FDA has granted approval for various drugs to treat COVID-19 patients, including remdesivir, nirmatrelvir, and ritonavir. As protease inhibitors, these medications are crucial in impeding virus replication. However, it is noteworthy that there are not yet any FDA-approved small molecules specifically designed to serve as viral entry inhibitors.

SARS-CoV-2 enters host cells via its spike (S) protein located on the viral surface [4,5,6]. The spike protein is a homotrimeric protein comprising S1 and S2 subunits. Within the S1 subunit, each monomer contains an N-terminal domain and a receptor-binding domain (RBD) [7,8]. Through direct binding, the virus utilizes the S1 RBD to attach to the angiotensin-converting enzyme 2 (ACE2) receptor on target cells [9,10], thereby initiating the first stage of viral entry. This interaction, therefore, represents an attractive target for antiviral therapeutics. Subsequently, fusion with the host cell membrane, facilitated by the S2 subunit, allows for the release of the viral contents into the cell cytoplasm. Upon ACE2 binding to S1, conformational changes and processing occur, enabling the fusion peptide (FP) in S2 to be inserted into the host cell membrane. The S2-heptad repeat (HR) regions HR1 and HR2 from each trimer subunit then undergo an anti-parallel refolding process, forming the six-helix bundle (6HB) [11]. This structural rearrangement brings the viral and cellular membranes into proximity, facilitating membrane fusion and the release of the viral genome into the cytoplasm. This step is critical for establishing productive infection, presenting another potential target for therapeutic intervention. 

In light of the intricate mechanism of SARS-CoV-2 entry into host cells described above, the blockage of SARS-CoV-2 into host cells is a logical and pivotal therapeutic approach. Firstly, as an early intervention, it disrupts a critical stage in the virus’s life cycle by preventing its entry into host cells. Secondly, targeting specific viral components during entry ensures a focused approach, minimizing off-target effects and enhancing the overall antiviral effectiveness [12,13]. Furthermore, given that many viruses rely on cellular endocytic mechanisms for infection [14,15], and that cells possess only a limited number of such pathways, blocking viral entry may disrupt these pathways, affecting numerous different viruses and significantly expanding our current array of antiviral strategies.

Several reports suggest that certain peptides and small molecules can reduce viral infection by blocking entry. Xia et al. have conducted research on peptide inhibitors, including EK1 and EK1C4, strategically designed to target the HR1 and HR2 domains within the spike protein. Notably, EK1 demonstrates effective inhibition against SARS-CoV-2 spike protein-mediated membrane fusion and pseudovirus infection in a dose-dependent manner [16]. Following this success, EK1C4, a lipopeptide derivative of EK1, was developed and confirmed to impede spike mediated cell–cell fusion [17]. Likewise, Zhu et al. designed IPB02, another peptide fusion inhibitor that effectively interferes with spike protein-mediated cell–cell fusion and pseudovirus infection [18]. Bojadzic, D. et al. discovered two novel druglike DRI small-molecule compounds, DRI-C23041 and DRI-C91005, which effectively hindered the interaction between human ACE2 (hACE2) receptors and the spike glycoprotein of SARS-CoV-2 in cell-free ELISA-type assays. These compounds display activity in the low micromolar range with IC_50_ values [19]. Using pseudotyped particle entry assays that conducted high-throughput screening, several small-molecule compounds sourced from approved drug libraries (Osimertinib, colforsin, ingenol, cepharanthine, abemaciclib, and trimipramine) were identified as spike-mediated entry inhibitors. The efficacy of their ability to reduce the cytopathic effect (CPE) caused by SARS-CoV-2 infection in Vero E6 cells was demonstrated, with IC_50_ values below 25 μmol/L [20]. The precise molecular mechanisms of action for these small-molecule entry inhibitors are not yet fully understood.

Given the pivotal role of SARS-CoV-2 spike protein interaction and ACE2 in facilitating virus entry, there is growing interest in using the obstruction of this interaction as a potential strategy to impede SARS-CoV-2 infection [8,21]. Acknowledging the significance of virus entry and the comparatively lower immunogenicity of the ACE2-binding surface on the RBD of the SARS-CoV-2 spike protein [22], our study adopts a focused approach targeting the RBD using computer-aided drug design (CADD) to identify potential compounds. Specifically, the compound structures identified in this study have the potential to serve as lead compounds for virus entry inhibition, offering promising avenues for COVID-19 treatment.

## 2. Results

### 2.1. A Framework for Constructing Anchors and Conducting Post-Screening Analysis

We initially attempted to identify potent inhibitors targeting the SARS-CoV-2 spike protein, utilizing docking algorithms, anchor construction, and post-screening analysis. The docking protein structure of the wild-type SARS-CoV-2 spike bound with the hACE2 model (PDB ID: 6M0J) was selected to generate SiMMap pivotal anchors in the receptor-binding site (RBS, as defined in the Section 4. A three-dimensional (3D) model depicts the anchor pockets and functional groups of the anchors (Figure 1). 

Three anchors (H1, V1 and V2) preferred to interact with N501 (Figure 1A), which required the nitrogen-containing or hydrophobic functional groups to maintain conformational stability. Of note, the compounds with a hydrophobic tendency, as well as the aromatic ring and heterocyclic moieties, demonstrated binding affinity with the V1 anchor (Figure 1B). The V2 anchor interacted with one small residue (G496), one polar uncharged amino acid (N501), and one hydrophobic residue (Y505) located at the RBS of the spike protein S1 subunit. This RBS binds to the cell receptor ACE2 in the region of aminopeptidase N [8]. Finally, the H1 anchor displayed a high tendency to bind to nitrogen-bonding moieties, interacting with two polar uncharged residues (Q498 and N501). A total of 279,156 NCI compounds underwent screening using iGEMDOCK molecular docking and the subsequent post-screening analysis. Following this, SiMMap anchors were generated from the SiMMap server, utilizing docking scores and the type of interactions observed between the top 1000 ranked compounds and the active sites. Each anchor comprises a binding pocket with associated interacting residues, moiety preference, and interaction type (E: electrostatic, H: hydrogen-bonding, or V: van der Waals forces). These 1000 compounds were re-ranked by SiMMap scores, and the top twenty compounds were selected as potential candidates and requested from the NCI. Subsequently, ten compounds (C1–C10, Appendix A) were available and identified for the in vitro cell-based inhibitory assay.

### 2.2. Discovery of Compounds C2, C8, and C10 as Selective Viral Entry Inhibitors

The MTS assay, a commonly used colorimetric method for evaluating cell viability and cytotoxicity, was employed to assess the cytotoxic effects of the ten potential compounds on BHK21-hACE2 cells. In Figure 2A, four compounds (C1, C3, C4, and C5) exhibited lower percentages of cell viability compared to the DMSO solvent control. However, none met the criteria for low relative cell viability (below 0.8). 

These results indicate that at a concentration of 100 μM, none of the ten compounds exhibited significant cytotoxic effects. Next, a wild-type SARS-CoV-2 pseudovirus inhibition assay on BHK21-hACE2 cells was performed to assess the efficacy of the ten compounds in virus entry. The pseudovirus contains luciferase genes, which can be detected through a luciferase assay when the pseudovirus infects cells. As depicted in Figure 2B, five of these ten compounds exhibited a significant reduction in the relative light unit (RLU) percentage compared to the DMSO-treated control group. Specifically, compounds C2, C6, C8, C9, and C10 decreased the RLU percentage values at concentrations of 25 μM compared to the DMSO solvent control, which decreased the percentage by 73.7%, 42.7%, 78.0%, 59.0%, and 76.7%, respectively. The precise RLU percentage values can be found in Appendix A. In contrast, C1 shows an increase in RLUs; however, this is beyond the scope of this article. We subsequently evaluated the top three compounds, namely C2, C8, and C10, which demonstrated inhibition rates exceeding 70% in the inhibition analysis, as potent inhibitors of wild-type SARS-CoV-2 entry. These compounds were further characterized by their half-maximal effective concentration (EC_50_), cytotoxicity concentration 50% (CC_50_), and selective index (SI). The CC_50_ was determined by the MTS assay at concentrations ranging from 100 μM to 500 μM, and the CC_50_ values of compounds C2, C8, and C10 all exceeded 500 μM (Figure 2C). To determine EC50, BHK21-hACE2 cells were infected with the pseudovirus (10,000 RLU/well) in the presence of individual compounds C2, C8, and C10 at concentrations of 6.25 μM, 12.5 μM, 25 μM, 50 μM, and 100 μM for 48 h. The EC_50_ values of C2, C8, and C10 are 8.8 μM, 6.7 μM, and 7.6 μM, respectively (Figure 2D). Detailed numerical data are presented in Appendix A. The calculated selectivity indices (SI) are >56.8 for C2, >74.6 for C8, and >65.8 for C10 (Table 1). These results indicate that C2, C8, and C10 were selective against wild-type SARS-CoV-2 infection. 

### 2.3. Characterization of Compound C8.2 as a Potent Viral Entry Inhibitor

Based on our initial findings, compounds C2, C8, and C10 exhibited promising inhibitory activity against wild-type SARS-CoV-2. We utilized the RDKit v1.5 software to generate structural fingerprints of these compounds for the purpose of performing an analog search in the NCI database. Nine analogs (Appendix A) were identified and the cell viability and viral entry inhibition assay were conducted. Compound C10 was not included in the analog search as it is a derivative of compound C8. Additionally, C2.3 and C2.4 exhibited low relative cell viability (below 0.8) and were therefore excluded from the antiviral efficacy test, while the remaining compounds showed no cytotoxicity (Figure 3A). In the viral entry inhibition assay, all compounds showed varying degrees of inhibition against viral infection, except C8.1 (Figure 3B). Notably, compound C8.2 demonstrated a potent inhibitory effect, blocking 97.3% of wild-type SARS-CoV-2 pseudovirus entry at a concentration of 25 μM compared to the DMSO control, highlighting its potential as a strong antiviral agent (Appendix A). Additionally, the CC_50,_ and EC_50_ of compound C8.2 were determined, The CC_50_ value was above 500 μM and the EC_50_ value was 5.9 μM (Figure 3C,D). Appendix A provide detailed numerical data corresponding to these figures. The SI ratio was >84.7 (Table 1), aligning with the findings from the pseudovirus inhibition assay. These results indicated that C8.2 was the most effective viral entry inhibitor among the tested compounds.

### 2.4. Evaluation of Compounds C2, C8, and Analogues against Omicron Variant (BA.1) Entry

Monoclonal antibodies, specifically engineered to target the spike protein of the SARS-CoV-2 virus, have been crucial in COVID-19 treatment. However, with the emergence of the Omicron variant and a high viral mutation rate, antibody treatments and cocktails face challenges in effectively combating the virus [23,24,25,26,27]. Our previous data have shown that most of the compounds, namely C2, C8, and their analogs, exhibit anti-SARS-CoV-2 activity during early-stage virus entry. We are interested in whether these compounds could effectively impede the entry of the Omicron variant virus into cells. In Figure 4A, ten of the eleven compounds significantly reduced BA.1 pseudovirus infection. The top four compounds, C2, C8, C10, and C8.2, demonstrated strong antiviral efficacy, with blocking percentages of 74.0%, 73.3%, 75.3%, and 86.0%, respectively. Further, the EC_50_ value of C8.2, exhibiting the highest efficacy, was determined, resulting in an EC_50_ of 9.4 μM (Figure 4B and Appendix A). The antiviral efficacy data for the remaining compounds are shown in Appendix A.

## 3. Discussion

In this study, we developed a strategy employing a spike inhibitor to disrupt the spike–ACE2 interaction. While ACE2 inhibitors are also viable interventions, we favored the spike inhibitor primarily due to concerns surrounding ACE2 inhibitors and their potential physiological consequences. ACE2 plays a crucial role in the renin–angiotensin–aldosterone system (RAAS), which is essential for regulating blood pressure and fluid balance. The inhibition of ACE2 may disrupt this balance, leading to RAAS dysregulation characterized by increased angiotensin II levels, promoting hypertension and fluid retention. Dysregulated RAAS activity is linked to cardiovascular diseases such as heart failure and stroke. Additionally, the observed ability of SARS-CoV-2 to induce the shedding of ACE2 receptors from the cell surface, resulting in a marked downregulation of ACE2 expression [28,29], intensifies these concerns. Given these factors, we contend that utilizing a spike inhibitor is a more favorable option as it directly targets the virus without compromising host cell function.

To better determine the precise mechanisms and interactions between the functional groups of our lead compounds C2, C8, C10, their analogs, and the residues of the SARS-CoV-2 spike protein, the possible interactions and structural analysis were discussed. Firstly, we found that C2 and its analogue C2.1 have a common backbone, namely glyoxal bis (2,4-dinitrophenyl-hydrazone), containing two para-directing nitro groups (Appendix A). The only difference between them is the side chain on the glyoxal, namely C2: 1,2,3-Butanetriol and C2.1: pentane-1,2,3,4-tetrol. The different lengths of the alkane side chain and the number of the hydroxyl group should influence the affinity between RBS and C2 and its analogue C2.1. From the experimental results of the C2 series, the inhibitory potencies between them indicate that C2 is higher than C2.1, followed by C2.2. It is worth noting that the omission of nitro functional moieties in C2.2 might decrease its affinity to binding to spike RBS; however, further validation is needed to confirm these observations. On the other hand, C8 and its analogues, including C10, have a 1,3,5-triazin derivative as their backbone (smile code: C1=CC=C(C=C1)CC(=O)NNC2=NC(=NC(=N2)NC3=CC=CC=C3)NNC(=O)C4=CC=NC=C4; IUPAC name: N’-[4-anilino-6-[2-(2-phenylacetyl)hydrazino]-1,3,5-triazin-2-yl]pyridine-4-carbohydrazide) (Appendix A). The most critical difference within the C8 series is the functional groups located in 4-anilino. We could divide the C8 series into three groups based on the directions of the functional group versus the nitrogen atom of 4-anilino linking to 1,3,5-triazin, namely the ortho, meta, and para directions. Only C8.2 has an ortho direction (2-methoxy group); C8, C9 and C10 have a meta direction (C8: 3-chloro, C9: 3-methyl, and C10: 3-methoxy group); and C8.1, C8.3, C8.4, and C8.5 possess a para direction (C8.1: 4-methyl propionate, C8.3: 4-chloro, C8.4: 4-methoxy, and C8.5: 4-nitro group). C8.2 showed highest potency against spike protein, which might provide more negative electrons towards spike protein and therefore increase the affinity.

Interaction profiles provide more insights into how these identified compounds interact with the spike protein. We could easily cluster the hits into different groups according to their interaction with the spike RBS. Specifically, the C2 series and C8 series were classified into two different sections of the heatmap diagram (Figure 5). The most potent three compounds validated by our designed bioassays are C8, C8.2, and C10, having interactive residues in common. 

Among those residues, the side chain of 500 T and 496 G demonstrated the highest affinity to bind to the three compounds. To check the possible interactions between the spike protein and the four potential compounds, C2, C8, C8.2, and C10, we analyzed these poses using PROTEINS PLUS(2022) software and the SiMMap(2010) server. From the C2 models, both Y505 and Q498 were shown in two different software (Figure 6A,E,I). Furthermore, taking C8, C8.2, and C10 together, we found that Q498 was the only residue shown in both software. In contrast, Y505 was only shown in C8.2 and C10, N501 was present in C8 and C8.2, and G496 was shown in C8 and C10. It is worth highlighting that both Q498 and Y505 have been proven to play an important role in binding to hACE2 [30,31,32,33,34,35]. 

Nitrogen-based heterocyclic chemistry plays a crucial role in small-molecular chemistry, demonstrating remarkable versatility and significance in various biological and pharmacological activities [36,37,38,39]. Specifically, our discovered inhibitors, namely C8 and its analogs (including C10), have a distinctive 1,3,5-triazine backbone and pyridine side chain. The 1,3,5-triazine, or s-triazine, is recognized as a versatile pharmacophore with lead biological activities, serving as the core framework for numerous antibacterial, antiviral, anticancer, and antifungal agents [40,41,42]. Recent studies evaluating s-triazine derivatives against SARS-CoV-2 have highlighted the antiviral efficacy of derivative 10a. Further investigations revealed its potential ability to inhibit the ATPase activity of the human helicase DDX3X [43]. Interestingly, our study found that compounds C8, C8.2, and C10, also featuring the s-triazine core structure, act as spike inhibitors rather than replication enzyme inhibitors. Therefore, future investigations may focus on exploring the potential helicase inhibition effects of compounds C8, C8.2, and C10. This could significantly enhance their antiviral efficacy and provide us with more potent therapeutic options against viral infections. 

Additionally, another nitrogen-based heterocyclic, the Pyridine scaffold, is associated with a wide range of biological activities, including antiviral, antibacterial, and antitubercular effects. Extensive research on pyridine derivatives from the years 2000 to 2020 has highlighted their significant antiviral activity against various viruses, including HIV, hepatitis C, hepatitis B, respiratory syncytial virus (RSV), and cytomegalovirus (CMV) [44]. Moreover, recent studies have shown promising results for the treatment of Coronavirus SARS-CoV-2 using pyridine derivatives. Terpyridine demonstrates potent binding efficacy against the SARS-CoV-2 spike glycoprotein [45], while bis-indolyl pyridines effectively hinder the infectivity of lentiviral vectors pseudotyped with spike glycoprotein. Further exploration of the interaction between bis-indolyl pyridines and spike RBD uncovered the formation of a hydrogen bond between Gly502 and one of the bis-indolyl pyridine ether groups [46]. Intriguingly, a similar hydrogen bond interaction is observed between C8.2 and Gly502. Gly502, identified through molecular docking studies of the S-RBD/ACE2 interface, forms highly stable hydrogen bonds with ACE2, particularly interacting with Lys353 for 57% of the simulation time [47]. These findings underscore the pivotal role of Gly502 in mediating the inhibitory effects of compounds against viral infections, highlighting the importance of targeting this interaction for the development of effective antiviral therapeutics. 

While this study demonstrates the inhibitory activity of compounds C2, C8, and their analogues against SARS-CoV-2, it is essential to acknowledge the limitations of our experiments. Due to the lack of a BSL-3 laboratory, the validation of viral infection was not conducted using authentic virus particles; instead, pseudovirus assays were employed. Additionally, our experimental design using the pseudovirus–luciferase system and docking energy data indirectly demonstrates that the compounds act as spike protein inhibitors. However, this is insufficient evidence to confirm their precise mechanism of action. Therefore, future studies should encompass an evaluation of the compounds’ antiviral activity against authentic viral strains and further biological assessments to comprehensively evaluate their antiviral potential.

## 4. Materials and Methods

### 4.1. Compound Library

The chemical compounds used in this study for virtual screening were collected from the National Cancer Institute (NCI). From the 2016 NCI database [48], 279,156 compounds were specifically chosen and then subjected to filtration based on Lipinski’s Rule of Five.

### 4.2. Molecular Docking

Molecular docking is a computer-based technique that attempts to find the most probable binding conformation of two molecules with minimized energy through maximized interactions. The structure of wild-type SARS-CoV-2 RBD (PDB ID:6M0J) was identified as the target protein for molecular docking from the Protein Data Bank [49]. Any residues at the intersection of the RBD domain and in a 10-angstrom area around ACE2 were selected as the target protein receptor-binding site (RBS). We employed iGEMDOCK v2.1 software, a docking program that computes pharmacological interactions by docking the compounds into the binding site of SARS-CoV-2 RBD. Based on the protein–protein interaction profiles and the ranking list generated using the energy-based docking scores, we picked out the top 1000 compounds.

### 4.3. Post-Screening Analysis

The top 1000 compounds with their binding poses were presented to the SiMMap server for post-screening analysis. SiMMap statistically infers the site-moiety map with anchors describing the interaction preferences between target protein-binding sites and compound moieties. The SiMMap server rescores a compound by combining the predicted docking energies from iGEMDOCK and the anchor score between the map and the compound. Based on the rearrangement scores by SiMMap, potential viral inhibitors were selected for further experimental investigations.

### 4.4. Cell Lines and Cell Culture

The BHK21-hACE2 cells were obtained from Dr. Yu Chia-Yi at the National Health Research Institutes in Miaoli, Taiwan. The cell is an engineered cell line derived from BHK21 cells, which stably express human ACE2 and serve as a cellular model for studying SARS-CoV-2 infection. The cells were cultured in Roswell Park Memorial Institute (RPMI) 1640 Medium (Gibco, Waltham, MA, USA), supplemented with 5% fetal bovine serum (FBS) with 1% penicillin/streptomycin, and maintained in a 5% CO_2_ atmosphere at 37 °C.

### 4.5. Cell Viability Assay

The MTS assay was employed to assess cell viability, as the manufacturer described. BHK21-hACE2 cells were seeded at a density of 1 × 10^4^ cells/well in a 96-well plate. The cells were treated with the tested compounds at the indicated dose for 48 h. DMSO-treated cells served as vehicle controls. After the period of exposure to the compounds, MTS reagent (G3580, Promega, Madison, WI, USA) was added to each well and incubated for an additional hour. Lastly, the absorbance at 490 nm was detected using SpectraMax^®^iD3 (Molecular Devices, San Jose, CA, USA). A blank control (RPMI1640 medium only) and vehicle controls (with DMSO defined as 100% cell survival) were included in every assay plate. Relative cell viability % = (A_treatment_ − A_blank_)/(A_control_ − A_blank_) × 100% (A: absorbance).

### 4.6. Generation of SARS-CoV-2 Spike Pseudovirus

We generated the wild-type SARS-CoV-2 pseudovirus to discover anti-SARS-CoV-2 entry inhibitors. 293T cells (7.5 × 10^5^ cells/well) were co-transfected with three plasmids to produce the pseudotyped viral particles, including an enveloped plasmid expressing the SARS-CoV-2 spike (pcDNA3.1-nCoV-SΔ18; RNAi Core, Academia Sinica, Taipei, Taiwan), a HIV-1 backbone plasmid expressing the packaging proteins (pCMV-Gag-Pol; RNAi Core, Academia Sinica, Taipei, Taiwan) and a transfer plasmid containing the firefly Luc reporter gene (pLAS3w.Fluc.Ppuro; RNAi Core, Academia Sinica, Taipei, Taiwan). The supernatant containing pseudovirus particles was harvested at 24 h, 36 h, and 48 h post-transfection, by centrifugation at 3000 rpm for 5 min and filtration through a 0.45 μm filter to remove cellular debris. Subsequently, a ten-fold concentration was achieved by utilizing a High-Speed Micro Refrigerated Centrifuge to concentrate the viral particles, thereby removing 90% of the virus-free medium. Finally, aliquots stored at −80 °C were used for entry inhibition experiments. The virus titers were determined by measuring the relative luminescence units (RLUs). For the Omicron pseudovirus, we purchased the Omicron (BA.1) lentivirus from the National Biotechnology Research Park, Academia Sinica, with order number A221170.

### 4.7. Pseudotyped SARS-CoV-2 Entry Inhibition Assay 

BHK21-hACE2 cells were seeded in a 24-well plate 24 h before infection at 5 × 10^4^ cells/well. Different compounds at indicated concentrations were mixed with the pseudovirus (10,000 RLU/well) at 37 °C for 30 min before the infection. The mixture was then transferred to the BHK21-hACE2 cells and incubation was continued for 48 h. For the vehicle control, the cells were treated with DMSO solvent. The luciferase activity was measured by the Bright-Glo™ assay (Promega, Madison, WI, USA) [50]. Relative Light Units % = (RLU_treatment_ − RLU_blank_)/(RLU_control_ − RLU_blank_) × 100%.

### 4.8. Statistical Analysis

Statistical analysis was performed using GraphPad Prism (version 10.2.0). Statistical significance was determined using one-way analysis of variance (ANOVA). All experiments were performed at least in triplicate. Statistical significance was shown as *: *p* < 0.05, **: *p* < 0.01, ***: *p* < 0.001, ****: *p* < 0.0001.

## 5. Conclusions

Our research provides an assessment of the viability of virtual screening utilizing SiMMap scores to identify anti-SARS-CoV-2 lead compounds. Initially, compounds C2, C8, and C10 exhibited the highest affinity to the wild-type spike RBD based on the bioassay results. Subsequent experimental assays on the derivatives of C2 and C8 led to the identification of compound C8.2 as the most efficacious against wild-type SARS-CoV-2. Furthermore, C8.2 also exhibited strong antiviral activity against the Omicron strain. The SiMMap server enabled desirable compound functional groups to be easily searched for and highlighted the interconnection between the RBD and compounds C2, C8, C10, and C8.2. Consequently, our findings support the utility of virtual screening methods in easily pinpointing potential anti-SARS-CoV-2 inhibitors that display significantly greater antiviral efficacy compared to existing strategies.

## Figures and Tables

**Figure 1 ijms-25-06105-f001:**
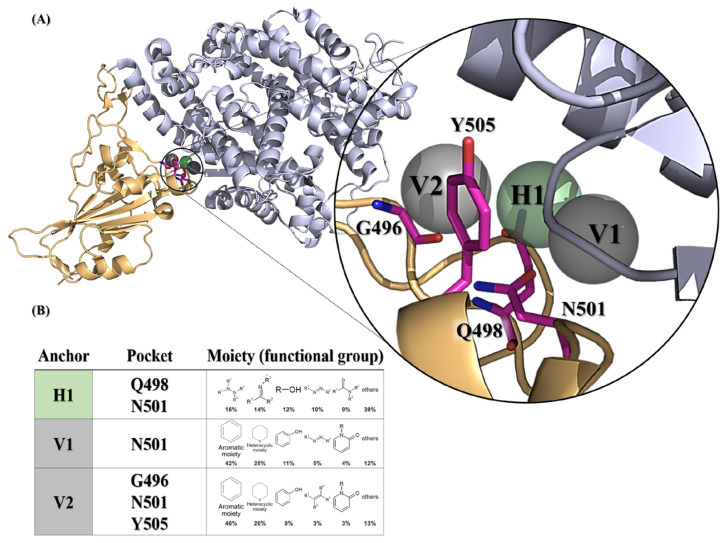
SARS-CoV-2 spike receptor-binding domain anchors produced from SiMMap. (**A**) The receptor-binding domain structure of the wild-type SARS-CoV-2 spike (chain E of 6M0J) is depicted as a cartoon and three anchors are shown as transparent spheres. (**B**) The binding pockets and moieties for each anchor. Each moiety of the anchor represents the functional group preference of the top-ranked compounds. H1 (shown as green areas) represents the hydrogen bonding forces, while V1 and V2 (shown as gray region) represent the van der Waals force. The corresponding binding pockets (residues) are represented as hot-pink sticks.

**Figure 2 ijms-25-06105-f002:**
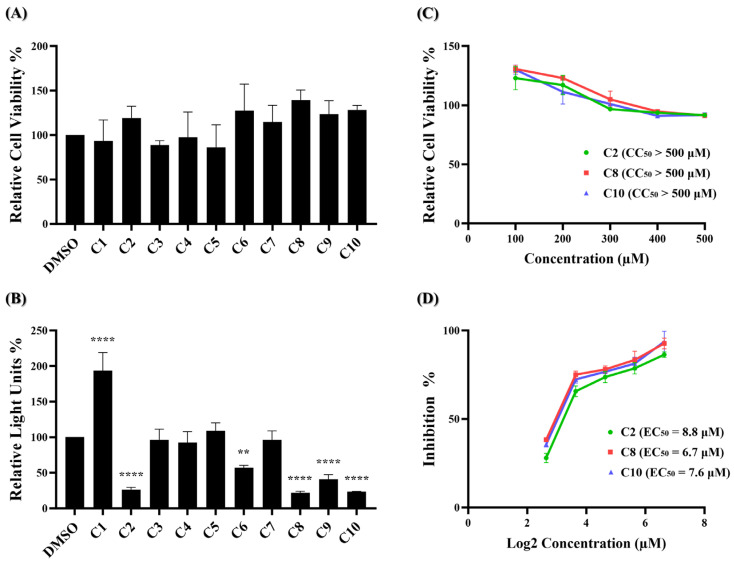
Identification of potent inhibitors through pseudovirus-based inhibition assay. (**A**) Each of the ten potential compounds was applied to the MTS assay at a concentration of 100 μM on BHK-hACE2 cells. Histograms represent the percentage of cell viability. Cells treated with solvent DMSO were used as a control and set to 100%. (**B**) Each of the ten potential compounds was applied to the pseudovirus-based inhibition assay on BHK21-hACE2 cells at a concentration of 25 μM. Histograms represent the percentage of RLUs and every column is compared to the DMSO control (DMSO, 100%). (**C**) CC_50_ and (**D**) EC_50_ values for C2, C8, and C10 were determined based on treatment with five concentrations. (CC_50_: 100 μM, 200 μM, 300 μM, 400 μM, 500 μM; EC_50_: 6.25 μM, 12.5 μM, 25 μM, 50 μM, 100 μM). CC_50_ was measured by the MTS cell viability assay. The y-axis indicates the percentage of cell viability. Data were normalized by the viability of the control group treated with DMSO. EC_50_ was measured by pseudovirus-based inhibition. The y-axis indicates the percentage of the antiviral effect. The x-axis is presented in log scale (base 2) of indicated concentrations. Statistical significance was shown as **: *p* < 0.01, ****: *p* < 0.0001.

**Figure 3 ijms-25-06105-f003:**
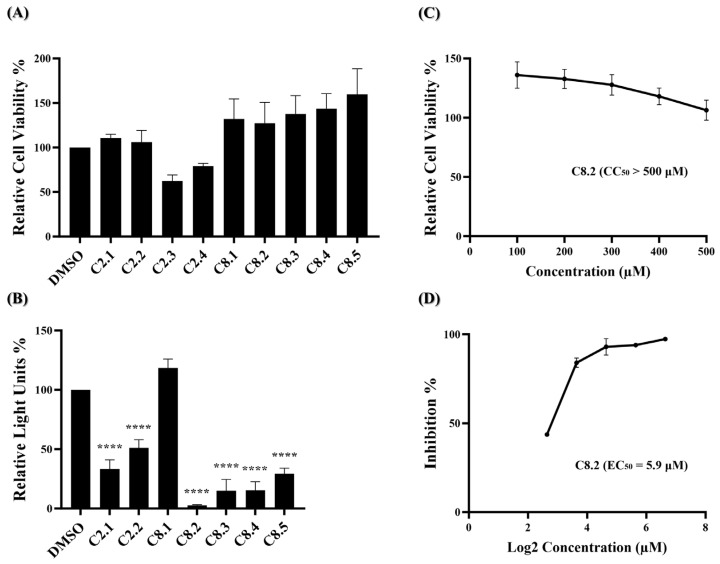
C2 and C8 analogs were assessed for antiviral efficacy. (**A**) Each of the analogs was applied to the MTS assay at a concentration of 100 μM. Histograms represent the percentage of cell viability. Cells treated in DMSO were set to 100%. (**B**) Each of the analogs was applied to the pseudovirus-based inhibitor screening assay at a concentration of 25 μM. Histograms represent the percentage of RLUs and every column is compared to the control (DMSO, 100%). (**C**) The CC_50_ of C8.2 was measured by the MTS assay and normalized by the viability of the control group treated with the DMSO solvent. (**D**) The EC_50_ of C8.2 was determined by the pseudovirus-based inhibition assay on BHK21-hACE2 cells. The x-axis is presented in log scale (base 2) of indicated concentrations. Statistical significance was shown as ****: *p* < 0.0001.

**Figure 4 ijms-25-06105-f004:**
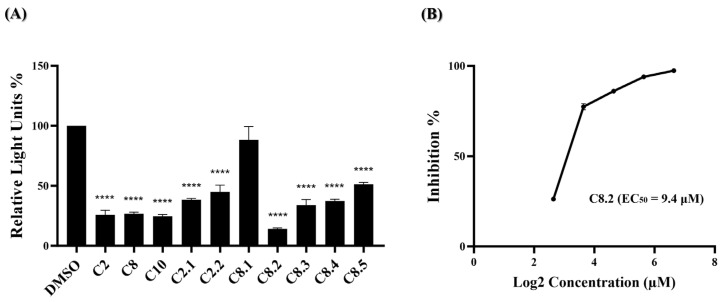
Anti-viral activity of potential compounds on BA.1 strain. (**A**) C2, C8, C10, and their analogues were applied to the pseudovirus-based inhibition assay using the BA.1 pseudovirus on BHK21-hACE2 cells, at a concentration of 25 μM. Histograms represent the percentage of RLUs and every column is compared to the control (DMSO, 100%). (**B**) The EC_50_ for C8.2 was determined by pseudovirus-based inhibition assays using the BA.1 pseudovirus. The y-axis indicates the percentage of antiviral effect. The x-axis is presented in log scale (base 2) of indicated concentrations. Statistical significance was shown as ****: *p* < 0.0001.

**Figure 5 ijms-25-06105-f005:**
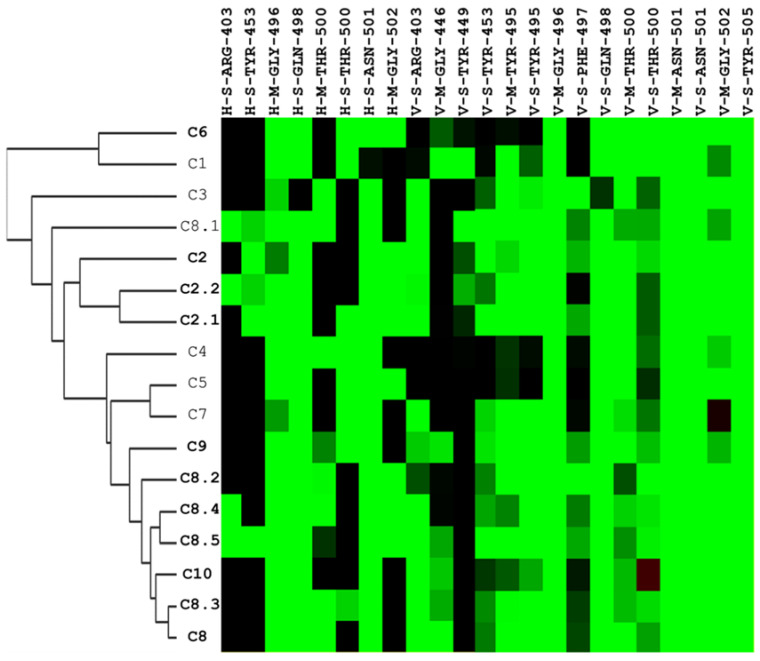
Hierarchical clustering analysis of interaction profiles using molecular docking poses of compounds identified to have potential inhibitory activity against the SARS-CoV-2 spike-binding domain. The hierarchical tree represents similarities among the compounds. On the y-axis, the compounds are listed, with those in bold indicating significant antiviral efficacy at a concentration of 25 μM, while the interactive residues are depicted on the x-axis. The first code of the interactive residue represents the forces between compounds and residues, namely E = electrostatic force, H = hydrogen bond force, and V = van der Waals force. The second code is the interaction in the main chain (M) or side chain (S). The third code represents the residue type and serial number of the SARS-CoV-2 spike protein. H and E interactions are represented in green when the energy ≤–2.5. V interactions are represented in green when the energy is <–4.

**Figure 6 ijms-25-06105-f006:**
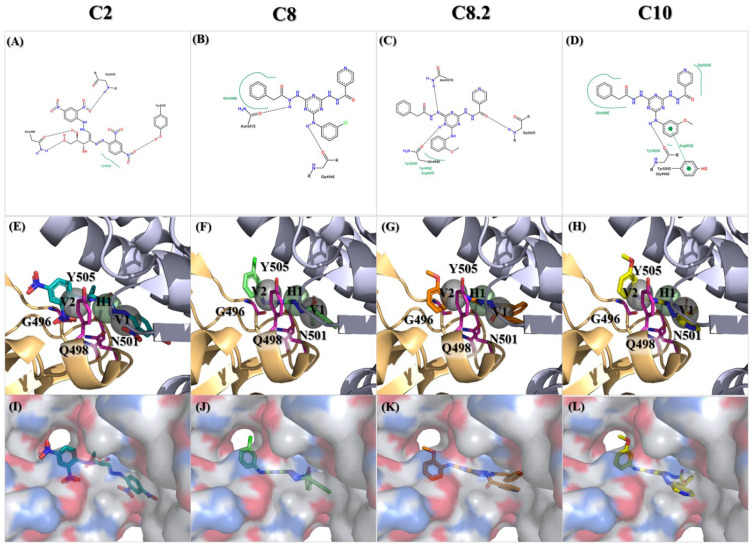
Molecular docking of the identified compounds with potential inhibitory activity against the SARS-CoV-2 spike-binding domain. (**A**–**L**) The molecular docking of compounds C2, C8, C8.2, and C10 with anchors in the binding domain of the SARS-CoV-2 spike is displayed in different visualization modes; (**A**–**D**) 2D plots indicate the interactions between docked compounds and the spike visualized with the PROTEINS PLUS server; (**E**–**H**) 3D images based on SiMMap results represent the interactions of docked compounds in the active site of the SARS-CoV-2 spike structure shown in the image. Anchors are shown as transparent spheres, interactive residues are shown as hot-pink sticks, and docked compounds are shown as deep green (C2), light green (C8), orange (C8.2), and yellow (C10)-colored sticks; (**I**–**L**) The interactions of docked compounds in the critical binding domain are depicted in the surface mode derived from Figure 6E–H using PyMOL v2.5.8 software.

**Table 1 ijms-25-06105-t001:** Summary of CC_50_, EC_50_, SI values, SiMMap scores and docking energies for compounds C2, C8, C10 and C8.2. The CC_50_ and EC_50_ values were determined from five-point dose response–curves using the MTS assay and the pseudovirus entry assay, respectively.

Compounds	CC_50_	EC_50_	SI	Score ^+^	Energy ^∞^
**C2**	>500 μM	8.8 μM	>56.8	3.409	−142.62
**C8**	>500 μM	6.7 μM	>74.6	3.401	−128.06
**C10**	>500 μM	7.6 μM	>65.8	3.394	−126.46
**C** **8.2**	>500 μM	5.9 μM	>84.7	3.383	−125.12

+ From SiMMap. ∞ From iGEMDOCK.

## Data Availability

The authors confirm that the data supporting the findings of this study are available within the article and its Appendix A.

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
