# Peer review of "Discovery of Novel Spike Inhibitors against SARS-CoV-2 Infection"

_ijms, 2024, doi:10.3390/ijms25116105_

Round 1

Reviewer 1 Report

Comments and Suggestions for Authors

This manuscript uses computer-aided drug screening to virtually screen 260000 compounds from the NCI compound library to discover novel backbone spike protein inhibitors. The spike protein receptor-binding domain was chosen as the target protein for our virtual screening process. Finally, the top 10 compounds with high scores were selected for antiviral activity evaluation in a pseudovirus system, and some pyridine scaffold compounds at the micromolar level were discovered. However, I don't think this article is enough to be directly published on IJMS, and he following comments for the authors should be considered: 

1. Figure 1B: Low resolution.

2. There is no standard deviation (SD) value for antiviral activity, and each concentration needs to be measured at least three times.

3. The article includes several comparisons with DMSO blank spaces; however, there is no experimental data on DMSO, and its purity may not be 100%, as shown in Figures 2A and 2B. Therefore, the accuracy of these data should be subject to debate.

4. Table 1 is redundant as it already exists in the figure.

5. When testing these compounds, positive control drugs such as namatevir or other S protein inhibitors should be included.

6. Merely relying on cellular activity against pseudo viruses is insufficient evidence to prove that this type of compound acts as an S protein inhibitor. Further experiments such as surface plasmon resonance (SPR) are required.

7. Were C8.1-C8.5 analogues of C8 obtained through rational drug design and synthesis based on its structure? If so, a brief description of their design concept, chemical synthesis process, and structural confirmation would be necessary.

8. SARS-CoV-2 has appeared for more than 4 years. At present, a lot of highly active inhibitors have been found, and the activity evaluation methods are also increasingly improved. Therefore, the activity of these compounds discovered by the author is relatively low, and their real antiviral activity has not been evaluated in real viruses.

Comments on the Quality of English Language

no comments

Reviewer 2 Report

Comments and Suggestions for Authors

This is a very interesting article.

Comments:

1.  Regarding the high viral mutation rate and the continuous production of new SARS-CoV-2 variants beyond Omicron, what should compounds’ characteristics be in order to function as “universal”? Otherwise, is there any estimation about the kinds of variants for which the actually tested compounds could present sufficient effectiveness?

2.  Limitations of any kind (circulating human proteins, co-existence of other microorganisms, etc) interfering with the molecular interactions between RBDs and receptors should be taken into account, investigated, or at least mentioned and described.

3.  What following research steps are suggested for the confirmation of the accessed knowledge, before leading to its application in treatment?

4.  The morphological quality (design, colours) of the graphs could improve.

Round 2

Reviewer 1 Report

Comments and Suggestions for Authors

Although Nirmatrelvir is not an invasion inhibitor, it can be used as a positive drug, at least it has an inhibitory effect on SARS-CoV-2; In addition, other S protein inhibitors can also serve as positive controls, but they are not included in this manuscript. 

Comments on the Quality of English Language

No comments